**Data Availability Statement:** The clinical results used in this study are based on findings in an RCT that we conducted, and published elsewhere (doi: 10.1016/j.jhsa.2018.09.015). All results on a group

# Volar locking plate versus external fixation for unstable dorsally displaced distal radius fractures–A 3-year cost-utility analysis

Jenny Saving[1,2☯], Emelie Heintz[3‡], Hans Pettersson[1‡], Anders Enocson[1,4‡], Cecilia Mellstrand Navarro[1,5☯]*

**1** Department of Clinical Science and Education, Södersjukhuset, Karolinska Institutet, Stockholm, Sweden, **2** Capio Artro Clinic, Stockholm, Sweden, **3** Department of Learning, Informatics, Management and Ethics (LIME), Karolinska Institutet, Stockholm, Sweden, **4** Department of Molecular Medicine and Surgery, Karolinska University Hospital, Karolinska Institutet, Stockholm, Sweden, **5** Department of Hand Surgery, Södersjukhuset, Stockholm, Sweden

☯ These authors contributed equally to this work.
‡ These authors also contributed equally to this work.
* cecilia.mellstrand-navarro@sll.se

## Abstract

### Aim

To investigate the cost-effectiveness of Volar Locking Plate (VLP) compared to External Fixation (EF) for unstable dorsally displaced distal radius fractures in a 3-year perspective.

### Methods

During 2009–2013, patients aged 50–74 years with an unstable dorsally displaced distal radius fracture were randomised to VLP or EF. Primary outcome was the incremental cost-effectiveness ratio (ICER) for VLP compared with EF. Data regarding health effects (Quality-adjusted life years, QALYs) was prospectively collected during the trial period until 3 years after surgery. Cost data was collected retrospectively for the same time period and included direct and indirect costs (production loss).

### Results

One hundred and thirteen patients (VLP n = 58, EF n = 55) had complete data until 3 years and were used in the analysis. At one year, the VLP group had a mean incremental cost of 878 euros and a gain of 0.020 QALYs compared with the EF group, rendering an ICER of 43 900 euros per QALY. At three years, the VLP group had a mean incremental cost of 1 082 euros and a negative incremental effect of -0.005 QALYs compared to the EF group, which means that VLP was dominated by EF. The probability that VLP was cost-effective compared to EF at three years, was lower than 50% independent of the willingness to pay per QALY.

level are available in the publication. However, individual clinical data are only available in the statistical files. The ethical permit does not allow us to make other than group data freely available online, but all files are fully available on request from the corresponding author, or from the Institution for Clinical Research and Education, Sodersjukhuset hospital, Karolinska Institutet, Sweden. We confirm that other researchers will be able to access the data in the same manner as the authors, since we are in posession of the statistical files without restrictions others than stated above. All other data is presented in its complete form in the submitted paper and its tables/figures/appendices. Data requests may be sent to the corresponding author (cecilia.mellstrand-navarro@sll.se) or the Karolinska Institutet telephone:0+46 8 524 875 04, E-mail:per.tornvall@ki.se

**Funding:** The project was conducted with financial support from the government founded ALF Region Stockholm, Sweden grants for research. The funder provided support in the form of salaries for authors JS, AE, and CMN, but did not have any additional role in the study design, data collection and analysis, decision to publish, or preparation of the manuscript. The specific roles of these authors are articulated in the 'author contributions' section.

**Competing interests:** Author JS is currently employed by a commercial company with no interest or role in conduction of this study. This does not alter our adherence to PLOS ONE policies on sharing data and materials.

## Conclusion

Three years after distal radius fracture surgery, VLP fixation resulted in higher costs and a smaller effect in QALYs compared to EF. Our results indicate that it is uncertain if VLP is a cost-effective treatment of unstable distal radius fractures compared to EF.

## Introduction

The incidence of surgical treatment of distal radius fractures has increased since the introduction of the volar locking plate (VLP) at the turn of the 21st century [1]. VLP has become the most commonly used surgical method, while the use of percutaneous methods, i.e. percutaneous pinning or external fixation (EF), has been reported to decrease [1–3]. There is little evidence to support that any surgical method yields superior clinical outcome as compared to others for treatment of distal radius fractures [4–6]. Other factors than final clinical outcome may therefore be allowed to influence treatment method choices. In a setting with limited health care resources, cost-effectiveness of different methods may be an important aspect to address in the choice of treatment, i.e. if the surgical methods have a reasonable incremental cost in relation to their effects. There is some evidence suggesting that VLP is not a cost-effective surgical technique when compared to percutaneous pinning [7,8]. To the best of our knowledge, health economic assessments of other treatment methods for distal radius fractures are largely lacking. No study has investigated cost- effectiveness of distal radius fracture surgery beyond a one-year perspective. The purpose of this study was to assess the cost-effectiveness of VLP versus EF for surgical treatment of patients 50–74 years old with a dorsally displaced distal radius fractures during the first 3 years after distal radius fracture surgery.

## Materials and methods

This study is a cost-utility analysis based on patients included in a previously published randomised controlled trial (RCT) comparing VLP with EF regarding functional outcome [6,9]. Patients eligible were 50–74 years of age with a distal radius fracture of >20 degrees dorsal angulation after a low energetic trauma presenting at a second-level trauma hospital in Stockholm, Sweden, during September 2009 to February 2013. Full inclusion and exclusion criteria are presented in Table 1. 140 patients were randomised through opening of sealed opaque envelopes to EF (Hoffman Compact T2, Stryker, Switzerland) or VLP fixation (2.4 mm Variable Angle LCP Two-Column Volar Distal Radius Plate, Synthes, Switzerland). Data regarding health effect was prospectively collected during the trial period. Cost data was collected retrospectively. The clinical 1-year and 3-year results [6,9] displayed no differences in Patient-Reported Outcome Measures (PROM) after the first 3 months. The analysis was conducted on an intention-to-treat basis. A power calculation was performed for detection of a 10-points difference in the main outcome Disability of the Arm, Shoulder and Hand (DASH) at one year follow-up of the initial RCT. A separate power calculation for the cost-utility outcomes was not conducted.

### Cost-effectiveness

This cost-utility analysis has been conducted using a health care perspective as well as using a broader perspective including production loss. The time horizons used are one and three years. The primary outcome was the incremental cost-effectiveness ratio (ICER) for VLP

**Table 1. Inclusion and exclusion criteria for patients with distal radius fracture for selection to a randomised controlled trial comparing volar locking plate and external fixation.**

| Inclusion criteria | Exclusion criteria |
|---|---|
| Patient age (50–74 years for women and 60–74 years for men) | Former disability of either wrist |
| Injury only after fall from standing height | Other concomitant injuries |
| Wrist radiography of >20 degree-dorsal angulation and/or >5 mm axial shortening (OTA class 23 A2, A3, C1, C2, C3) | Rheumatoid arthritis or other severe joint disorder |
| Good knowledge of written and spoken Swedish | Dementia or Pfeiffer score*<5 |
| Fracture diagnosed within 72 hours from injury | Drug or alcohol abuse, or psychiatric disorder |
| Patient resident within the catchment area of the hospital | Dependency in activities of daily living |
| | Medical condition contraindicating general anaesthesia |

*Adapted from Pfeiffer, E. A short portable mental status questionnaire for the assessment of organic brain deficit in elderly patients. J Am Geriatric Soc. 1975;23:433–441.

compared with EF using a health care perspective including production loss. The ICER was defined as the difference in mean total cost per patient divided by the difference in mean Quality-adjusted-life years (QALY) per patient, expressed as the incremental cost per gained QALY for VLP compared with EF. If the mean difference in QALYs was negative and the mean difference in total cost positive, no ICER was calculated, as VLP then was considered to be dominated by EF. If the mean difference in total cost was negative and the mean difference in QALYs positive, no ICER was calculated, as VLP then was considered to dominate EF. Both costs and QALYs were in line with national guidelines discounted at a discount rate of three percent [10].

## Costs

Total costs per patient were calculated by first identifying and estimating the resource use associated with each surgical method and then valuing each resource using the unit costs presented in Table 2. The direct costs and indirect costs during the first year and up to 3 years for each treatment were summed up.

## Unit costs

All unit costs are presented in Table 2. Unit costs for operating theatre including staff were derived from a report by the Swedish Agency for Health Technology Assessment and Assessment of Social Services (SBU) [11]. Costs regarding in-and outpatient care including emergency ward visits were collected from the diagnose-related group (DRG [12]) financial reimbursement system used at the hospital. Costs for drug usage were calculated from prices defined in FASS [13] (a compilation from the pharmaceutical industry with information about drugs used in Sweden) for a Defined Daily Dose as defined in the Drug Registry of the Swedish National Board of Health and Welfare [14]. Unit costs for reoperations were calculated based on estimations by the study group regarding surgical time and material usage. All costs above are considered as direct costs. Indirect costs consisted of production loss due to sick leave after the surgery. The unit cost regarding production loss per day was derived from Statistics Sweden [15], using the mean income for adults 20–74 years plus taxes and social services fee. Costs from 2016 were converted to 2018 years prices using a 2% mark-up for every year. All costs are presented in euros converted from Swedish kronor (SEK) with an exchange rate of 0.0978.

**Table 2. Unit costs used in a cost-utility analysis comparing volar locking plate and external fixation in patients with distal radius fractures.**

| | Unit | Cost (Euro) | Reference |
|---|---|---|---|
| **Direct costs** | **Costs for primary surgery** | | |
| | Volar locking plate implant including intraoperative antibiotics, dressings and cast^ | 441.4 | Manufacturers price list |
| | External fixation implant and dressings^^ | 122.3 | Manufacturers price list |
| | Operation theatre minute including fixed equipment + overhead costs per minute | 2.69 | SBU¤ |
| | Operation assistant per minute | 0.73 | SBU¤ |
| | Surgical nurse per minute | 1.08 | SBU¤ |
| | Anaesthetic nurse per minute | 1.08 | SBU¤ |
| | Anaesthesist per minute | 2.15 | SBU¤ |
| | Orthopaedic surgeon per minute | 2.15 | SBU¤ |
| | **Costs for reoperations** | | |
| | Carpal ligament release | 597.6 | * |
| | Tendon transfer | 1049.9 | ** |
| | Volar locking plate fixation | 1731.4 | *** |
| | Volar locking plate extraction/screw extraction | 895.4 | **** |
| | Soft tissue surgery (fasciotomy, scar release, secondary suture, wound debridement | 856.7 | ***** |
| | **Costs for hospital care** | | |
| | Emergency visit | 375.9 | DRG |
| | Outpatient visit | 153.8 | DRG |
| | Day of inpatient care | 449.9 | DRG |
| | Occupational therapy, visit | 108.4 | DRG |
| | **Costs for X-ray** | 51.8 | DRG |
| | **Costs for Drugs** | | |
| | Antibiotics, 1-day use, regular dose | 1.53 | FASS Drug registry |
| | Paracetamol, 1-day use, regular dose | 0.41 | FASS, Drug registry |
| | Opioids, 1-day use, regular dose | 1.91 | FASS, Drug registry |
| | Non-steroid anti-inflammatory drugs, 1-day use, regular dose | 0.25 | FASS, Drug registry |
| | Neuroleptics | 1.40 | FASS, Drug registry |
| **Indirect costs** | Production loss per day | 179.46 | SCB |

^2.4-mm Variable Angle LCP Two-Column Volar Distal Radius Plate, Synthes, Switzerland.

^^Hoffman Compact T2, Stryker, Switzerland.

¤ **Mellstrand Navarro C, Brolund A, Ekholm C, Heintz E, Hoxha Ekstrom E, Josefsson PO, Leander L, Nordstrom P, Ziden L, Stenstrom K.** Treatment of radius or ulna fractures in the elderly: A systematic review covering effectiveness, safety, economic aspects and current practice. *PLoS One 2019;14–3:e0214362.*

*15 min surgical time, 40 min preparation time and 60 min postoperative time in the operation theatre. Operation assistant and surgical nurse attending all time, orthopaedic surgeon attending during surgical time, 10 min before surgery and 10 min after surgery. Dressings.

**45 min surgical time, 40 min preparation time and 60 min postoperative time in the operating theatre. Operation assistant, surgical nurse and anaesthetic nurse attending all time, orthopaedic surgeon attending during surgical time, 10 min before surgery and 10 min after surgery and anaesthesist attending 45 min. Dressings.

***70 min surgical time, 40 min preparation time and 60 min postoperative time in the operating theatre. Operation assistant, surgical nurse and anaesthetic nurse attending all time, orthopaedic surgeon attending during surgical time, 10 min before surgery and 10 min after surgery and anaesthesist attending 45 min. Volar locking plate implant including dressings, cast and one dose of antibiotics. One x-ray.

****25 min surgical time, 40 min preparation time and 60 min postoperative time in the operating theatre. Operation assistant, surgical nurse and anaesthetic nurse attending all time, orthopaedic surgeon attending during surgical time, 10 min before surgery and 10 min after surgery and anaesthesist attending 45 min. Dressings.

*****20 min surgical time, 40 min preparation time and 60 min postoperative time in the operating theatre. Operation assistant, surgical nurse and anaesthetic nurse attending all time, orthopaedic surgeon attending during surgical time, 10 min before surgery and 10 min after surgery and anaesthesist attending 45 min. Dressings.

SBU, the Swedish Agency for Health Technology Assessment and Assessment of Social Services.

DRG, Diagnose-related group financial reimbursement system used at the hospital.

FASS, a compilation from the pharmaceutical industry with information about drugs used in Sweden.

Drug registry, a registry held by the Swedish National Board of Health and Welfare.

## Resource use

All resources needed for each treatment method were identified by the research group. Resource use data for surgical time for the primary surgery was derived from prospectively inserted data in the surgery software system used at the hospital (Orbit [16]). Inpatient and outpatient visits for diagnoses related to the initial injury and any possible related complication (International Classifications of Disease, ICD-10 codes [17] specified in Appendix) were retrieved at an individual level as registry data from the Swedish National Board of Health and Welfare. Drug usage was defined as prescription of antibiotics and analgesics (Anatomical Therapeutic Chemical Classification [18], ATC drug codes specified in Appendix) collected as registry data from the Swedish National Board of Health and Welfare. Data regarding sick leave for diagnoses related to the initial injury and any possible related complication (ICD-10 codes specified in Appendix) were collected as registry data from the Swedish Social Insurance Agency. Any reoperations were detected by search of patient records, and/or registry data retrieval from the Swedish National Board of Health and Welfare regarding surgical procedures related to any possible related complication (NOMESCO classification for surgical procedures codes [19] specified in Appendix). Estimations of resource use were performed by the study group for occupational therapy and x-rays since no complete registry or study protocol source was available. The time frame for all resource use was set to from the date of the injury to the date of the 3-year follow-up.

## Effectiveness

Effectiveness of treatment was estimated using Quality of Life Adjusted Life Years (QALYs) [20,21]. QALYs are a composite measure of survival and Health related Quality of Life, HRQoL. One QALY can be interpreted as the equivalent of one year in full health. The QALYs following each treatment during the study period was calculated on an individual level using the area under the curve (AUC) approach [22]. QALYs for each time interval were calculated by taking the average of the HRQoL at two adjacent time points multiplied with the time in years spent in each time interval. The QALYs gained at 1 year and at 3 years were summarized and an average for each time period was calculated. The HRQoL of the patients was estimated using EuroQol 5 dimensions, EQ-5D-3L [23] and was reported by trial participants at baseline, 2 weeks, 6 weeks, 3 months, 1 year and 3 years postoperatively. EQ-5D-3L is a measure of health status and consists of a questionnaire with five questions and a visual analogue scale (EQ-VAS) [23]. The five questions each represent a dimension of health; mobility, self-care, usual activities, pain/discomfort and anxiety/depression. Each question has three response levels and can be combined into a health profile of five digits [23], which was converted into a health state value using a value set from the United Kingdom (UK) [24].

## Statistical analyses

Data were analysed using SPSS version 26. A complete case analysis was conducted to avoid violating the assumption that data was missing at random, i.e. no imputations were made and only participants with complete data were analysed.

Categorical data was compared with Chi-square test. Normality was tested with Shapiro-Wilks test for all continuous variables. For normally distributed variables independent Student's t-test was used. Skewed distributed data was compared with Mann-Whitney U-test. Kolmogorov-Smirnov's test and Kruskal-Wallis´ test were used to confirm statistical significance for non-parametric comparisons. The level of statistical significance was set to $p < 0.05$ in two-sided tests. Linear regression was used to adjust mean differential QALYs at 1 year and 3 years for imbalance between groups in EQ-5D-3L index scores at baseline [25]. Health state values

(the EQ-5D-3L index scores in this study) at baseline (before treatment) is often invariably imbalanced between trial arms and it is recommended that the comparisons between treatments are adjusted for this imbalance as it otherwise will contribute to a difference in QALYs that is not an effect of the treatments [25]. Therefore, the difference in mean QALYs between VLP and EF was adjusted for differences in EQ-5D-3L index scores between VLP and EF at baseline (before surgery).

The non-parametric bootstrapping approach with replacement [26] was used to determine the level of sampling uncertainty around the ICER. The bootstrap was performed as a resampling from the original sample to create 1000 random samples. In each bootstrap sample, 58 individuals among the VLP patients and 55 individuals among the EF patients were randomly selected with equal probability and with replacement after each individual selection. To adjust for baseline differences in EQ-5D-3L index scores between the groups [25], we calculated the adjusted differential QALYs (VLP = intervention, EF = control) in each sample.

1000 estimates of incremental costs and effects were generated. The bootstrap is presented in a cost-effectiveness plane [26]. From the bootstrap a cost-effectiveness acceptability curve (CEAC) was derived at 1 year and 3 years, to express the probability that VLP is cost-effective in comparison to EF for a range of thresholds for willingness to pay (WTP) per gained QALY [26].

A threshold of 35000 euros was chosen as maximum WTP per gained QALY, which approximates the 30000 UK pounds sterling used by the National Institute for Health and Clinical Excellence (NICE) in the UK [27].

### Ethics

The conduction of this study was approved by the Regional board for ethical vetting, Stockholm, Sweden, ref nr 2008/1908-31/4, 2009/715-31/2, 2012/2201-32, 2012/1363-32, 2016/2207-32. The collection of data analysed in this trial was recorded at clinicaltrials.gov (NCT 01034943, NCT01035359).

## Results

Of the 140 patients randomised, 6 dropped out before the first year and 16 thereafter, leaving 118 patients for the 3-year follow-up. There were no missing data regarding resource use. Of the 118 patients, five had not filled in all EQ-5D-3L questionnaires and were excluded, leaving 113 patients (VLP n = 58, EF n = 55) for the cost-effectiveness analysis. From the EF group, four patients were converted to VLP intraoperatively and four received a volar plate within the first two weeks after primary surgery, but they were still evaluated within the EF group. Baseline characteristics are presented in Table 3.

### Resource utilization and costs

Resource utilization is presented in Table 4. All costs are presented in Table 5 and Fig 1. The mean total cost was significantly higher for the VLP group compared with the EF group at 1 year (mean difference; MD: 878 euros, p = 0.006). Mean total cost increased for both groups until the 3-year follow-up, and VLP costs were still significantly higher (MD: 1 082 euros, p = 0.012).

### Health-related quality of life

EQ-5D-3L index scores are presented in Table 6 and Fig 2. At 2 and 6 weeks, the VLP group had statistically significant better EQ-5D-3L index scores than the EF group, but differences

**Table 3. Baseline characteristics of study population in a cost-utility analysis comparing volar locking plate and external fixation in patients with distal radius fractures.**

|  | Volar locking plate (n = 58) | External fixation (n = 55) | P-value |
|---|---|---|---|
| Women (%) | 51 (88%) | 53 (96%) | 0.163* |
| Age, mean (SD) | 63 (6.3) | 63 (6.7) | 0.460** |
| Injury to dominant hand | 21 (36.2%) | 29 (53%) | 0.090* |
| AO-class**** |  |  |  |
| - A2 | 3 (5.5%) | 3 (5.2%) | 0.898*** |
| - A3 | 19 (34.5%) | 15 (25.9%) |  |
| - C1 | 32 (58.2%) | 31 (53.4%) |  |
| - C2 | 2 (3.6%) | 4 (6.9%) |  |
| - C3 | 2 (3.6%) | 2 (3.4%) |  |

*Chi-square test.

**Student's t-test.

***Fisher's exact test.

****Müller ME, Nazarian S, Koch P, The Comprehensive Classification of Fractures of Long Bones, Springer Verlag, Berlin, Heidelberg, 1990.

SD, Standard Deviation.

did not remain at later follow-up time points. Mean EQ-5D-3L index scores improved continuously between all timepoints but was still lower than pre-injury levels at 3 years. Mean total QALYs during the first year was 0.814 in the VLP group and 0.787 in the EF group (p = 0.236) (Table 7). After adjustments for baseline differences in EQ-5D-3L index scores between the groups, the difference in mean total QALYs was 0.020 (p = 0.344) in favor of the VLP group. At 3 years, mean total QALYs was 2.530 in the VLP group and 2.518 in the EF group (p = 0.852). The adjusted mean difference was 0.005 (p = 0.932) in favor of the EF group (Table 7).

## Cost-utility analysis

From a health care perspective, the ICER at 1 year was 22 100 euros per QALY for VLP fixation compared to EF (Table 7). When including production loss, the ICER increased to 43 900 euros per QALY. At 3 years, VLP resulted in higher costs and a smaller effect in QALYs than EF, independent of whether production loss was included or not. This means that VLP was dominated by EF in the longer time horizon. The bootstrap analyses of the estimates including production loss are presented in cost-effectiveness planes (Fig 3). The scatterplot covers all four quadrants indicating uncertainty about whether or not VLP is cost-effective and at what value it is cost-effective compared to EF. The Cost Effectiveness Acceptability curves (CEAC) in Figs 4 and 5 summarize the probability of VLP being cost-effective compared to EF at one and three years respectively. At a willingness to pay threshold of 35 000 euros per QALY, the probability that VLP is cost-effective compared to EF is around 50% at 1 year and 40% at 3 years. At 3 years, the probability that VLP is cost-effective does not exceed 50% independent of the willingness to pay per QALY.

## Discussion

The purpose of this study was to assess the cost-effectiveness of VLP compared to EF. The study shows that at 3 years, VLP patients had higher costs and a smaller effect (although not statistically significant) in QALYs compared to EF patients independent of the perspective

**Table 4. Resource utilization used in a cost-utility analysis comparing volar locking plate and external fixation in patients with distal radius fractures.**

| Resource Utilization | | | | | | | | | | |
|---|---|---|---|---|---|---|---|---|---|---|
| Unit | Volar locking plate (n = 58) | | | | External fixation (n = 55) | | | | Diff (mean) | P-value |
| | Mean (SD) | Min, Max | Median | Missing (%) | Mean (SD) | Min, Max | Median | Missing (%) | | |
| **Primary surgery** | | | | | | | | | | |
| Time in operating theatre (min) | 70 (18) | 36–113 | 68 | 0 | 43 (24) | 19–135 | 35 | 0 | 27 | <0.001 |
| Preparing time + postoperative time in operating theatre (min) | 40 + 60 | | | | 40 + 60 | | | | | |
| **Reoperations** | | | | | | | | | | |
| Carpal ligament release, | | | | | | | | | | |
| - 1st year | 0.02 (0.13) | 0–1 | 0.0 | 0 | 0.02 (0.14) | 0–1 | 0.0 | 0 | -0.001 | 0.970 |
| - 3 years | 0.02 (0.13) | 0–1 | 0.0 | 0 | 0.02 (0.14) | 0–1 | 0.0 | 0 | -0.001 | 0.970 |
| Tendon transfer | | | | | | | | | | |
| - 1st year | 0.02 (0.13) | 0–1 | 0.0 | 0 | 0.0 | | 0.0 | 0 | 0.017 | 0.330 |
| - 3 years | 0.02 (0.13) | 0–1 | 0.0 | 0 | 0.0 | | 0.0 | 0 | 0.017 | 0.330 |
| Volar locking plate fixation | | | | | | | | | | |
| - 1st year | 0.02 (0.13) | 0–1 | 0.0 | 0 | 0.05 (0.23) | 0–1 | 0.0 | 0 | -0.037 | 0.286 |
| - 3 years | 0.02 (0.13) | 0–1 | 0.0 | 0 | 0.05 (0.23) | 0–1 | 0.0 | 0 | -0.037 | 0.286 |
| Plate extraction | | | | | | | | | | |
| - 1st year | 0.09 (0.28) | 0–1 | 0.0 | 0 | 0.04 (0.19) | 0–1 | 0.0 | 0 | 0.05 | 0.274 |
| - 3 years | 0.17 (0.38) | 0–1 | 0.0 | 0 | 0.04 (0.19) | 0–1 | 0.0 | 0 | 0.13 | 0.019 |
| Soft tissue surgery (fasciotomy, scar release, secondary suture, wound debridement | | | | | | | | | | |
| - 1st year | 0.07 (0.53) | 0–4 | 0.0 | 0 | 0.05 (0.05) | 0–1 | 0.0 | 0 | 0.014 | 0.298 |
| - 3 years | 0.09 (0.54) | 0–4 | 0.0 | 0 | 0.05 (0.05) | 0–1 | 0.0 | 0 | 0.032 | 0.623 |
| **Hospital care** | | | | | | | | | | |
| Emergency visits | 1 | | | | 1 | | | | | |
| Outpatient visits | | | | | | | | | | |
| - 1st year | 5.1 | 4–9 | 5 | 0 | 5.7 (1.5) | 2–11 | 5 | 0 | -0.6 | 0.005 |
| - 3 years | 6.4 | 4–14 | 6 | 0 | 6.8 (6.0) | 3–13 | 6 | 0 | -0.4 | 0.165 |
| Inpatient care days | | | | | | | | | | |
| - 1st year | 0.5 (1.9) | 0–10 | 0 | 0 | 0.2 (0.7) | 0–4 | 0 | 0 | 0.3 | 0.974 |
| - 3 years | 0.5 (1.9) | 0–10 | 0 | 0 | 0.2 (0.7) | 0–4 | 0 | 0 | 0.3 | 0.974 |
| Occupational therapy visits | | | | | | | | | | |
| - 1st year | 4 | 4 | 4 | 0 | 5 | 5 | 5 | 0 | -1 | Not relevant** |
| - 3 years | 4 | 4 | 4 | 0 | 5 | 5 | 5 | 0 | -1 | |
| **X-ray** | 2 | 2 | 2 | 0 | 2 | 2 | 2 | 0 | 0 | Not relevant** |
| **Drugs, daily doses** | | | | | | | | | | |
| Antibiotics | | | | | | | | | | |
| - 1st year | 1.0 (4.3) | 0–25.0 | 0.0 | 0 | 6.4 (20.0) | 0–113.0 | 0.0 | 0 | -5.4 | 0.099 |

(*Continued*)

**Table 4.** (Continued)

| Unit | Volar locking plate (n = 58) | | | | External fixation (n = 55) | | | | Diff (mean) | P-value |
|---|---|---|---|---|---|---|---|---|---|---|
| | Mean (SD) | Min, Max | Median | Missing (%) | Mean (SD) | Min, Max | Median | Missing (%) | | |
| - 3 years | 3.4 (8.4) | 0–37.5 | 0.0 | 0 | 8.3 (23.5) | 0–123.8 | 0.0 | 0 | -4.9 | 0.746 |
| Paracetamol | | | | | | | | | | |
| - 1st year | 21.4 (55.9) | 0–291.7 | 0.0 | 0 | 16.1 (35.9) | 0–200.0 | 0.0 | 0 | 5.3 | 0.952 |
| - 3 years | 44.0 (110.0) | 0–641.8 | 0.0 | | 61.5 (143.8) | 0–678.1 | 0.0 | 0 | -17.5 | 0.789 |
| Opioids | | | | | | | | | | |
| - 1st year | 32.0 (32.9) | 0–206.0 | 25.7 | 0 | 27.9 (25.5) | 0–117.0 | 23.3 | 0 | 4.2 | 0.539 |
| - 3 years | 39.4 (49.1) | 0–299.1 | 32.7 | 0 | 39.6 (78.5) | 0–575.2 | 23.3 | 0 | -0.2 | 0.495 |
| Non-steroid anti-inflammatory drugs | | | | | | | | | | |
| - 1st year | 16.8 (67.8) | 0–423.3 | 0.0 | 0 | 8.0 (26.7) | 0–160.0 | 0.0 | 0 | 8.8 | 0.843 |
| - 3 years | 49.3 (173.1) | 0–1212.5 | 0.0 | 0 | 29.5 (65.7) | 0–320.0 | 0.0 | 0 | 19.8 | 0.669 |
| Neuroleptic drugs | | | | | | | | | | |
| - 1st year | 1.3 (7.3) | 0–50.0 | 0 | 0 | 0 | 0 | 0 | 0 | 1.3 | 0.167 |
| - 3 years | 1.6 (8.3) | 0–58.3 | 0 | 0 | 0 | 0 | 0 | 0 | 1.6 | 0.089 |
| **Sick leave (days)** | | | | | | | | | | |
| - 1st year | 19.9 (46) | 0–259 | 0 | 0 | 17.5 (32) | 0–114 | 0 | 0 | 2.4 | 0.650 |
| - 3 years | 20.5 (49 | 0–291 | 0 | 0 | 17.5 (32) | 0–114 | 0 | 0 | 3.0 | 0.650 |

*Student's t-test.

** Estimations of resource use were performed by the study group since no complete registry or study protocol source was available.

used, indicating that VLP is not cost-effective compared to EF. At 1 year, the VLP group had gained more QALYs than the EF group, and the incremental total cost per QALY gained for the health care perspective (excluding production loss) was below the threshold recommended by NICE. However, when including production loss, the threshold was exceeded. Between 1 year and 3 years, VLP patients increased their costs more than EF patients and EF patients increased their EQ-5D-3L index scores more than VLP patients. The statistical analyses displayed a high level of uncertainty surrounding the ICER, which implies that further studies are needed to support our findings.

There are no previous cost-utility studies comparing VLP with EF, but there are some studies comparing VLP with percutaneous pinning. Tubeuf et al [8] found a statistically significant incremental cost of 815 euros (converted from UK pounds sterling) after one year for VLP compared with percutaneous pinning. As VLP patients had a smaller gain in QALYs (0.008) than in our study, the resulting ICER was higher (100 295 euros per QALY). However, they did not investigate patients beyond the first year. Karantana et al [7] presented a study comparing VLP with percutaneous pinning and optional EF (11/64 patients) and showed a statistically significant incremental cost of 801 euros (converted from UK pounds sterling) after 1 year. They also presented a smaller gain in QALYs (0.0178) for VLP patients than our study, resulting in an ICER of 44 990 euros per QALY for the VLP group in comparison with the percutaneous pinning group.

**Table 5. Costs in euros used in a cost-utility analysis comparing volar locking plate and external fixation in patients with distal radius fractures.**

| | Volar locking plate Mean (SD) | External fixation Mean (SD) | Diff (mean) | p-value* |
|---|---|---|---|---|
| Implant | 441.4 (0) | 145.4 (86.6) | 295.9 | <0.001 |
| Operation theatre | 456.8 (47.5) | 385.4 (65.3) | 71.4 | <0.001 |
| Operation staff | 780.2 (89.0) | 646.5 (604.7) | 133.7 | <0.001 |
| **Total cost for primary surgery** | 1678.3 (136.5) | 1177.4 (260.8) | 500.9 | <0.001 |
| **Reoperations** | | | | |
| Carpal ligament release | | | | |
| - 1st year | 10.3 (78.5) | 10.9 (80.6) | -0.6 | 0.970 |
| - 3 years | 10.3 (78.4) | 10.9 (80.6) | -0.6 | 0.970 |
| Tendon transfer | | | | |
| - 1st year | 18.1 (137.9) | 0 | 18.1 | 0.330 |
| - 3 years | 18.1 (137.9) | 0 | 18.1 | 0.330 |
| Volar locking plate fixation | | | | |
| - 1st year | 29.9 (227.4) | 94.4 (396.8) | -64.6 | 0.286 |
| - 3 years | 29.9 (227.4) | 94.4 (396.8) | -64.6 | 0.286 |
| Volar locking plate extraction | | | | |
| - 1st year | 77.2 (253.5) | 32.6 (169.1) | 44.6 | 0.274 |
| - 3 years | 152.1 (336.2) | 32.6 (169.1) | 119.6 | 0.023 |
| Soft tissue surgery | | | | |
| - 1st year | 59.1 (450.0) | 46.7 (196.3) | 12.4 | 0.298 |
| - 3 years | 73.4 (461.2) | 46.7 (196.3) | 26.7 | 0.606 |
| All reoperations | | | | |
| - 1st year | 194.5 (608.6) | 184.6 (516.1) | 9.9 | 0.962 |
| - 3 years | 283.8 (726.7) | 184.6 (516.1) | 99.2 | 0.358 |
| **Hospital care** | | | | |
| Outpatient care including primary emergency visit | | | | |
| - 1st year | 1161.1 (140.3) | 1254.2 (228.7) | -93.2 | 0.005 |
| - 3 years | 1347.4 (255.2) | 1408.3 (284.7) | -60.8 | 0.101 |
| Inpatient care | | | | |
| 1st year | 232.7 (831.0) | 106.3 (311.7) | 126.4 | 0.974 |
| 3 years | 232.7 (831.0) | 106.3 (311.7) | 126.4 | 0.974 |
| Occupational therapy | | | | Not relevant** |
| - 1st year (4432/5540) | 433.4 | 541.8 | -108.4 | |
| - 3 years (4432/5540) | 433.4 | 541.8 | -108.4 | |
| **X-ray** | | | | Not relevant** |
| - 1st year | 103.7 | 103.7 | 0 | |
| - 3 years | 103.7 | 103.7 | 0 | |
| **Drugs** | | | | |
| Antibiotics | | | | |
| - 1st year | 1.55 (6.58) | 9.79 (30.51) | -8.2 | 0.099 |
| - 3 years | 5.06 (12.38) | 12.56 (35.58) | -7.5 | 0.743 |
| Paracetamol | | | | |
| - 1st year | 8.79 (22.96) | 6.61 (14.75) | 2.2 | 0.952 |
| - 3 years | 17.63 (44.14) | 24.43 (56.91) | -6.8 | 0.751 |
| Opioids | | | | |
| - 1st year | 61.12 (62.81) | 53.20 (48.58) | 7.9 | 0.539 |
| - 3 years | 74.45 (91.95) | 74.46 (143.58) | -0.01 | 0.482 |
| Non-steroid anti-inflammatory drugs | | | | |
| - 1st year | 4.11 (16.58) | 1.96 (6.53) | 2.1 | 0.843 |
| - 3 years | 11.72 (41.22) | 7.02 (15.65) | 4.7 | 0.677 |

(*Continued*)

**Table 5.** (Continued)

|  | Volar locking plate Mean (SD) | External fixation Mean (SD) | Diff (mean) | p-value* |
|---|---|---|---|---|
| Neuroleptics |  |  |  |  |
| - 1st year | 1.81 (10.19) | 0 | 1.8 | 0.167 |
| - 3 years | 2.19 (11.60) | 0 | 2.2 | 0.089 |
| All drugs |  |  |  |  |
| - 1st year | 77.38 (94.28) | 71.56 (62.58) | 5.8 | 0.968 |
| - 3 years | 111.05 (161.66) | 118.47 (193.55) | -7.4 | 0.859 |
| **Total direct costs** |  |  |  |  |
| - 1st year | 3881 (1439) | 3440 (897) | 442 | **<0.001** |
| - 3 years | 4190 (1640) | 3641 (921) | 550 | **<0.001** |
| **Indirect costs** |  |  |  |  |
| Sick leave |  |  |  |  |
| - 1st year | 3575 (8261) | 3138 (5677) | 436 | 0.650 |
| - 3 years | 3671 (8785) | 3138 (5677) | 533 | 0.650 |
| **Total cost (direct and indirect costs)** |  |  |  |  |
| - 1st year | 7456 (8329) | 6578 (5745) | 878 | **0.006** |
| - 3 years | 7861 (9011) | 6778 (5733) | 1082 | **0.012** |

*Mann-Whitney U-test.

**Estimations of resource use were performed by the study group since no complete registry or study protocol source was available.

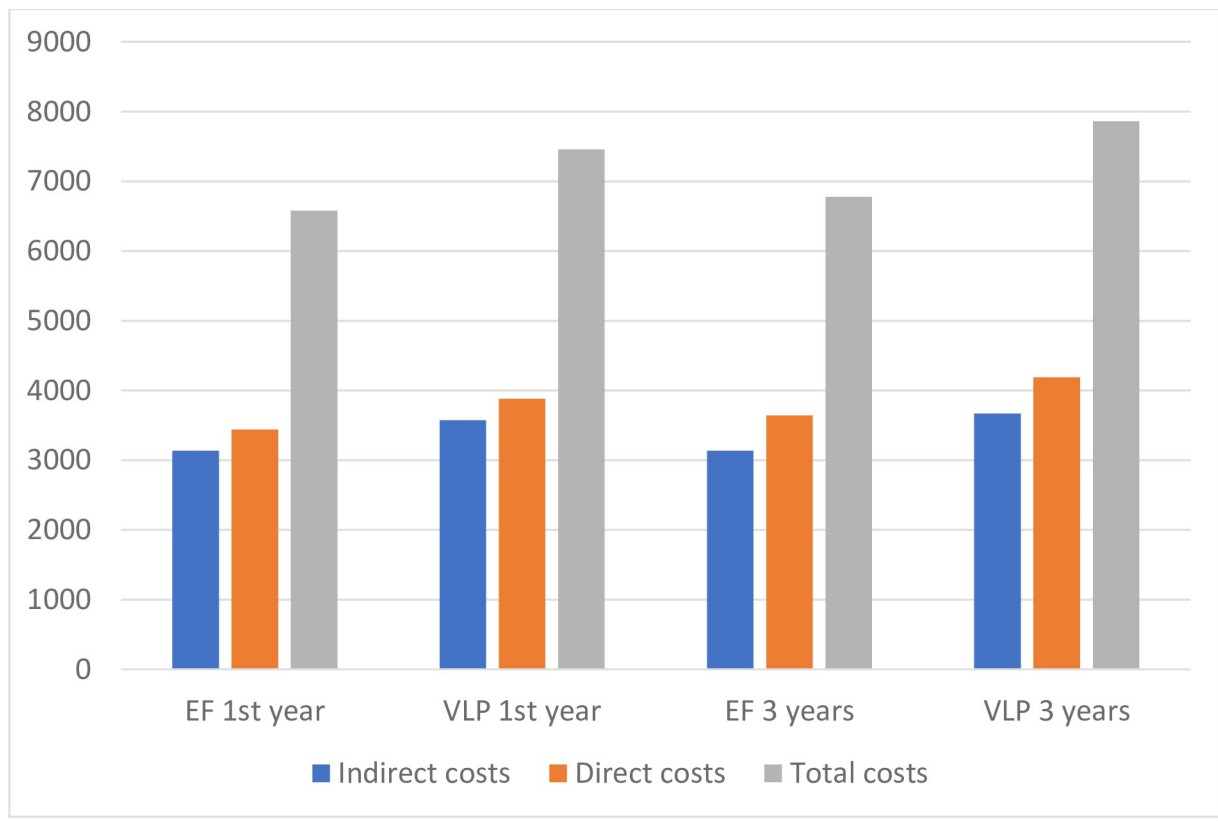

**Fig 1. Mean costs for external fixation (EF) patients and volar locking plate (VPL) patients one and three years after distal radius fracture surgery.**

**Table 6. Mean EQ-5D-3L index scores at pre-injury, baseline and follow-up points after distal radius fracture surgery with volar locking plate and external fixation.**

| EQ-5D-3L index score | Volar locking plate Mean (SD) | External fixation Mean (SD) | p-value* |
|---|---|---|---|
| Pre-injury | 0.970 (0.076) | 0.936 (0.129) | 0.104 |
| Baseline | 0.502 (0.278) | 0.458 (0.317) | 0.652 |
| 2-week follow-up | 0.705 (0.197) | 0.624 (0.217) | **0.018** |
| 6-week follow-up | 0.757 (0.189) | 0.674 (0.208) | **0.009** |
| 3-month follow-up | 0.820 (0.112) | 0.777 (0.176) | 0.158 |
| 1-year follow-up | 0.877 (0.189) | 0.889 (0.132) | 0.766 |
| 3-year follow-up | 0.917 (0.132) | 0.921 (0.131) | 0.852 |

*Mann-Whitney U-test.

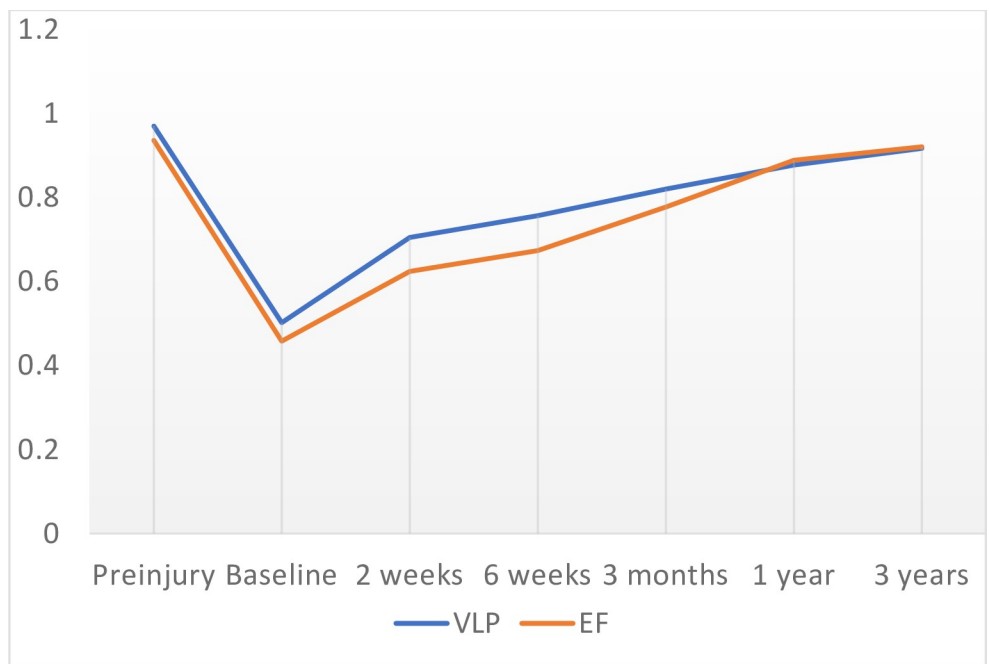

**Fig 2. Mean EQ-5D-3L index scores at preinjury, baseline and follow-up points after surgery with volar locking plate (VLP) and external fixation (EF).**

**Table 7. Cost-utility analysis for volar locking plate fixation (VLP) compared to external fixation (EF) after distal radius fracture surgery.**

| | Costs (Euro) 1st year | QALYs 1st year | Cost per QALY gained 1st year | Costs (Euro) 3 years | QALYs at 3 years | Cost per QALY gained at 3 years |
|---|---|---|---|---|---|---|
| **Health care perspective** | | | | | | |
| VLP | 3881 | 0.814 | **22 100** | 4190 | 2.5302 | **Dominated** |
| EF | 3440 | 0.787 | | 3641 | 2.5181 | |
| Difference | **442** | 0.020* | | **550** | -0.005* | |
| Health care perspective plus production loss | | | | | | |
| VLP | 7456 | 0.814 | **43 900** | 7861 | 2.5302 | **Dominated** |
| EF | 6578 | 0.787 | | 6778 | 2.5181 | |
| Difference | 878 | 0.020* | | 1082 | -0.005* | |

*Adjusted for baseline differences.

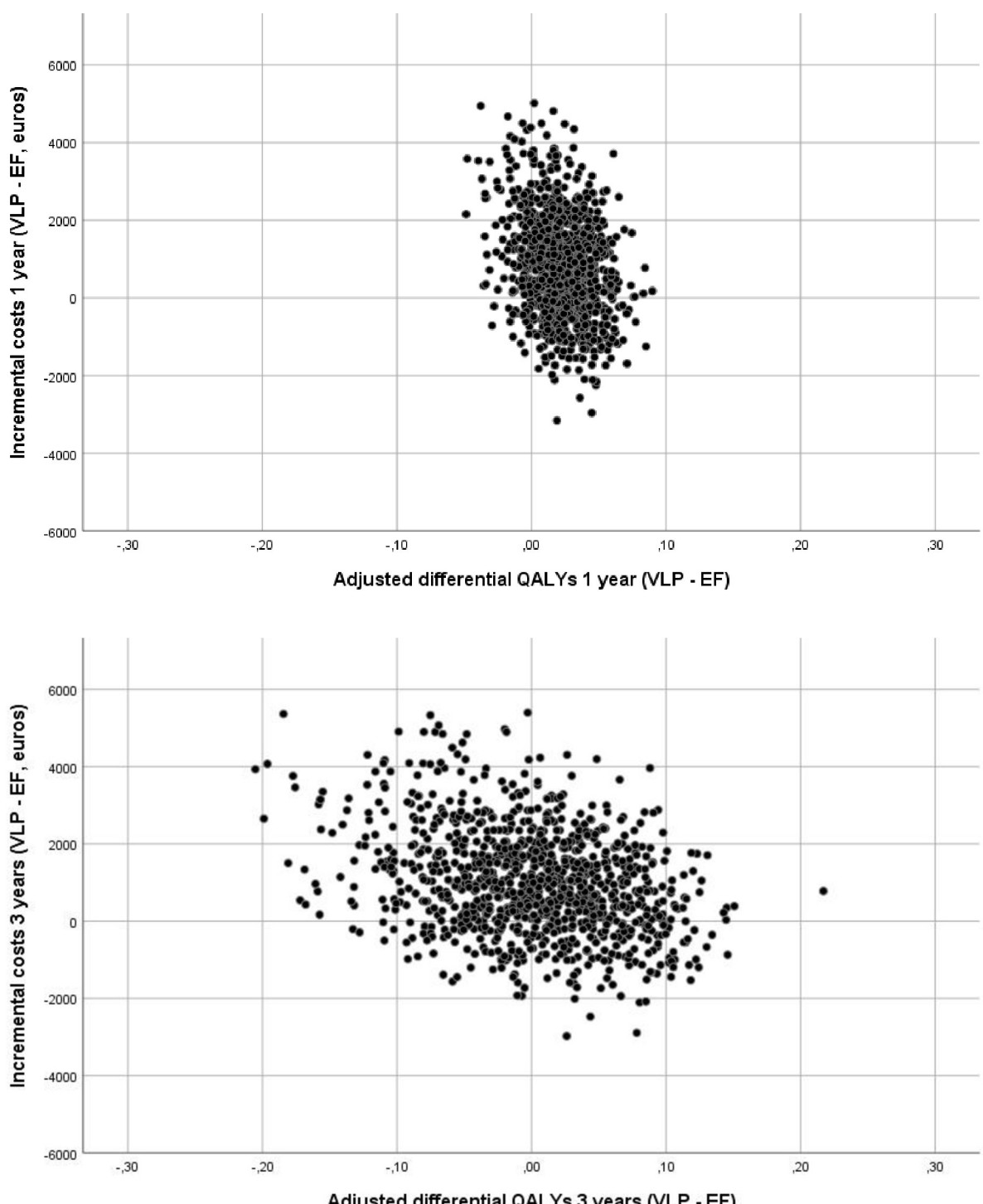

**Fig 3. Scatterplots of 1 000 samples of bootstrapped differences in mean costs and quality adjusted life years (QALYs) (adjusted for baseline difference in EQ-5D-3L index scores) over one year and three years after volar locking plate (VLP) compared to external fixation (EF), in cost-effectiveness planes.**

Differences in EQ-5D-3L index scores and resulting QALYs were very small in the studies of Tubeuf [8] and Karantana [7], which is in accordance with the findings in our study. Even small differences in total costs render large differences in ICER due to small differences in

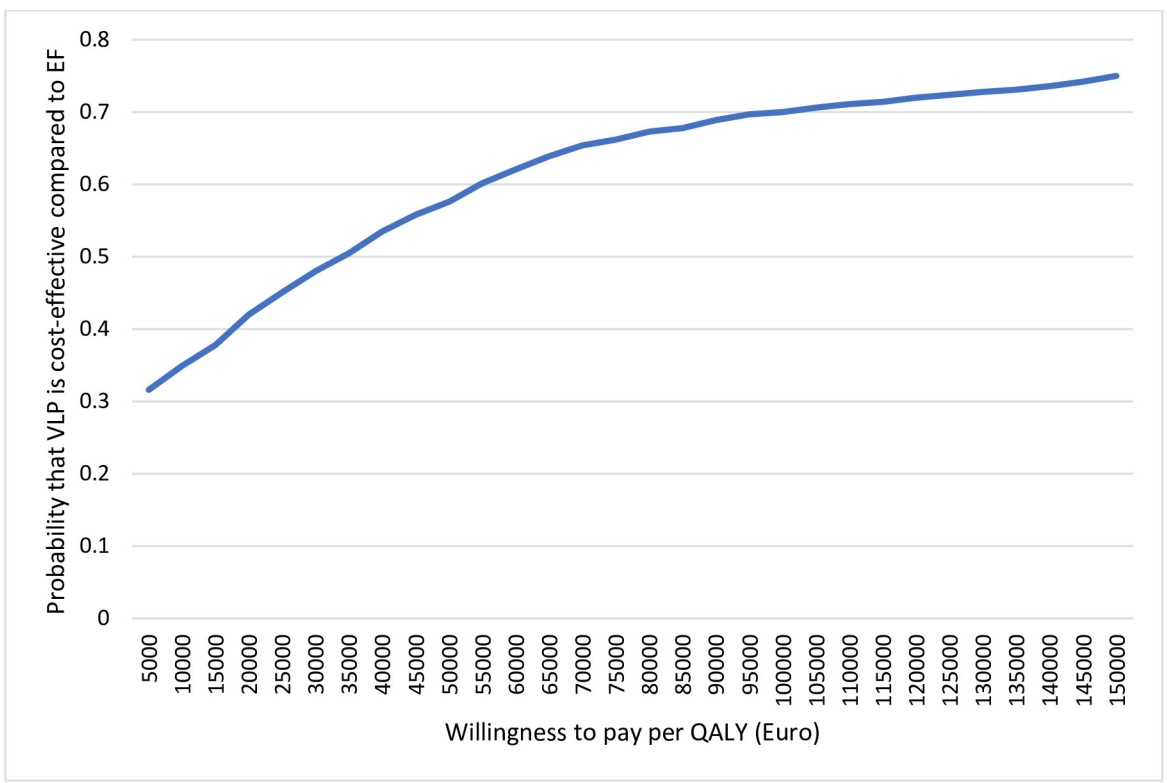

**Fig 4. Cost-effectiveness acceptability cure (CEAC) representing the probability of the cost-effectiveness of treatment using a volar locking plate (VLP) compared with external fixation (EF) at different willingness to pay (WTP) thresholds at one year after distal radius fracture surgery.**

QALYs. As VLP was associated with higher costs, VLP would still not be considered cost-effective even if there were no differences in QALYs.

The major strength of the present study is the relatively long follow-up period as treatment-related costs still occur after the first year, and HRQoL continues to improve. Another strength is that the study is conducted within the scope of a randomised trial, thus decreasing the risk of an impact on the results of potential biases. One strength is also that we have used registry data from the Swedish National Board of Health and Welfare, thereby capturing any resource use occurring at other hospitals or care providers. The use of registry data is also a limitation as we searched for ICD-10 codes and drug prescriptions that we assumed could be associated with the distal radius fracture, possibly rendering an overestimation of outpatient visits and drug usage. Another limitation is that we could not, in the retrospective perspective, evaluate the resource use of occupational therapy and x-rays and therefore had to make an estimation. Moreover, there was no data on primary care or nurse visits. Lastly, the study population is relatively small, thus limiting the power of detecting small differences between groups.

In conclusion, VLP fixation was associated with higher costs and resulted in fewer QALYs gained compared to EF at 3 years after distal radius fracture surgery., At this time horizon, the probability of VLP being cost-effective as compared to EF did not exceed 50% when including production loss, independent of the willingness to pay per QALY when adaption a perspective including production loss. Thus, our results indicate that it is uncertain if VLP is a cost-effective treatment of unstable distal radius fractures compared to EF.

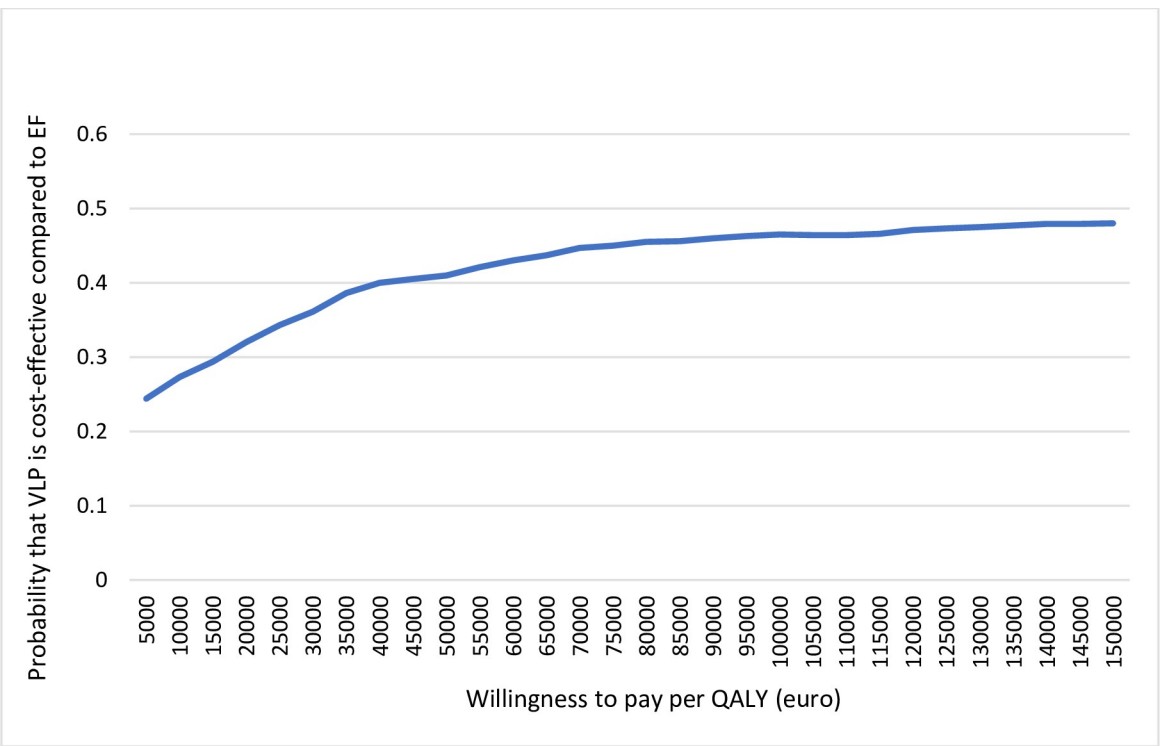

**Fig 5. Cost-effectiveness acceptability cure representing the probability of the cost-effectiveness of treatment using a volar locking plate (VLP) compared with external fixation (EF) at different willingness-to-pay thresholds at three years after distal radius fracture surgery.**

## Supporting information

**S1 File. A CHEERS checklist.**
(PDF)

## Acknowledgments

We wish to acknowledge research nurses Tina Levander and Elisabeth Skogman for invaluable help.

## Author Contributions

**Conceptualization:** Jenny Saving, Emelie Heintz, Hans Pettersson, Anders Enocson, Cecilia Mellstrand Navarro.

**Data curation:** Jenny Saving, Emelie Heintz, Hans Pettersson, Cecilia Mellstrand Navarro.

**Formal analysis:** Jenny Saving, Emelie Heintz, Hans Pettersson, Cecilia Mellstrand Navarro.

**Funding acquisition:** Anders Enocson, Cecilia Mellstrand Navarro.

**Investigation:** Jenny Saving, Emelie Heintz, Hans Pettersson, Cecilia Mellstrand Navarro.

**Methodology:** Jenny Saving, Emelie Heintz, Hans Pettersson, Anders Enocson, Cecilia Mellstrand Navarro.

**Project administration:** Jenny Saving, Cecilia Mellstrand Navarro.

**Resources:** Jenny Saving, Hans Pettersson, Anders Enocson, Cecilia Mellstrand Navarro.

**Software:** Jenny Saving, Emelie Heintz, Hans Pettersson, Cecilia Mellstrand Navarro.

**Supervision:** Emelie Heintz, Hans Pettersson, Anders Enocson, Cecilia Mellstrand Navarro.

**Validation:** Jenny Saving, Emelie Heintz, Hans Pettersson, Cecilia Mellstrand Navarro.

**Visualization:** Jenny Saving, Emelie Heintz, Hans Pettersson, Cecilia Mellstrand Navarro.

**Writing – original draft:** Jenny Saving, Emelie Heintz, Hans Pettersson, Anders Enocson, Cecilia Mellstrand Navarro.

**Writing – review & editing:** Jenny Saving, Emelie Heintz, Hans Pettersson, Anders Enocson, Cecilia Mellstrand Navarro.

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
