## [Decision Letter · Decision Letter 0]

10 Jun 2020

PONE-D-20-09180

Volar Locking Plate versus External Fixation for Unstable Dorsally Displaced Distal Radius Fractures – A 3-year Cost-Utility Analysis

PLOS ONE

Dear Dr. Mellstrand Navarro,

Thank you for submitting your manuscript to PLOS ONE. After careful consideration, we feel that it has merit but does not fully meet PLOS ONE’s publication criteria as it currently stands. Therefore, we invite you to submit a revised version of the manuscript that addresses the points raised during the review process.

We look forward to receiving your revised manuscript.

Kind regards,

Daniel Ribeiro

Academic Editor

PLOS ONE

Journal Requirements:

The authors have declared that no competing interests exist.

We note that one or more of the authors are employed by a commercial company: Capio Artro Clinic.

Additional Editor Comments (if provided):

Thank you for your submission. The reviewers provided excellent feedback to improve the quality and clarity of your reporting. They have also identified important areas that need revision.

I would like to highlight the following:

- Please ensure you adhere to the Consolidated Health Economic Evaluation Reporting Standards (CHEERS) statement (this was also raised by Reviewer #2).

- Table 1: needs formatting, it is not formatted as table.

Reviewers' comments:

Reviewer's Responses to Questions

**Comments to the Author**

1. Is the manuscript technically sound, and do the data support the conclusions?

Reviewer #1: Yes

Reviewer #2: Yes

2. Has the statistical analysis been performed appropriately and rigorously? 

Reviewer #1: Yes

Reviewer #2: Yes

3. Have the authors made all data underlying the findings in their manuscript fully available?

Reviewer #1: Yes

Reviewer #2: No

4. Is the manuscript presented in an intelligible fashion and written in standard English?

Reviewer #1: Yes

Reviewer #2: Yes

5. Review Comments to the Author

Reviewer #1: I will focus on methods and reporting. The abstract is well written and balanced. The use of the ICER is appropriate as far as I can tell. Methods are appropriate. Graphs are informative.

Major

1) the sample size, for which the authors cannot do much. But they can tone down the certainty of their findings and conclusions, especialyl in the abstract.

2) Linked to that there is no power calculation. Was there for the original study? It is worth saying that there was, to detect what (if there was), and then go on to say that you are underpowered to detect anything but very large effects. A post-hoc power calculation is meaningless.

2) missing data - why is a complete case approach selected? Why don't the authors use multiple imputation, for example?

Minor

1) Introduction was a bit short and did not give a comprehensive picture of the issue.

2) I appreciate this is an old trial and the information is avaialble elsewhere, but the authors could expand a bit on the randomisation process at least since it is quite important.

Reviewer #2: This is a generally well-conducted study, and the authors' interpretations and conclusions are supported by the results, although should perhaps be softened somewhat given the wide uncertainty evident in the cost-effectiveness results.

The reporting is generally clear, although some minor grammatical issues should be addressed by careful language/copy-editing throughout.

I have a few further concerns that should be addressed before the manuscript is suitable for publication:

1. The cost-perspective is probably best described as a partial societal perspective, as the only non-healthcare costs considered are production loss due to sick leave (i.e. many other societal costs, such as other out-of-pocket patient costs, caregiver time, travel, etc. are not included)

2. Reporting a health care sector perspective in additional to the (partial) societal perspective would be useful (e.g. the dual reference cases recommended by the 2nd panel on cost-effectiveness in health & medicine). This would likely also give more precise results, given how much the productivity losses contribute to the overall uncertainty

3. Would be good to see a completed CHEERS checklist to ensure the manuscript conforms to reporting guidelines for health economic evaluations

4. The authors have stated that 'All relevant data are within the manuscript and its Supporting Information files.' If the complete data have been uploaded as Supporting Information, I do not seem to have access to them as a reviewer, and there is no list of supporting information included

5. There are a lot of figures & tables included, some of which could probably be moved to an Appendix (e.g. Table 4) or combined (e.g. Figures 4 & 5)

6. PLOS authors have the option to publish the peer review history of their article (what does this mean?). If published, this will include your full peer review and any attached files.

Reviewer #1: No

Reviewer #2: No

---

## [Author Response · Author response to Decision Letter 0]

31 Jul 2020

Dear PLOS One,

Please consider our resubmission of PONE-D-20-09180

Volar Locking Plate versus External Fixation for Unstable Dorsally Displaced Distal Radius Fractures – A 3-year Cost-Utility Analysis

PLOS ONE

All our comments are provided below in italics. We hope that our efforts are sufficient. Do not hesitate to contact us for further clarifications or modifications of our submission.

Sincerely yours, 

Cecilia Mellstrand Navarro and co-authors

We have checked all style requirements to the best of our abilities.

2. We note that one or more of the authors are employed by a commercial company: Capio Artro Clinic.

a. Please provide an amended Funding Statement declaring this commercial affiliation

We have added a “funding statement declaration” to the cover letter clarifying the affiliation of author JS, and the roles of our funder.

We have added this information in the cover letter.

Additional Editor Comments (if provided):

Thank you for your submission. The reviewers provided excellent feedback to improve the quality and clarity of your reporting. They have also identified important areas that need revision.

I would like to highlight the following:

- Please ensure you adhere to the Consolidated Health Economic Evaluation Reporting Standards (CHEERS) statement (this was also raised by Reviewer #2).

A Cheers statement has been added as supporting information S1.

- Table 1: needs formatting, it is not formatted as table.

Table 1 has been formatted as a table.

Reviewers' comments:

Reviewer's Responses to Questions

Comments to the Author

1. Is the manuscript technically sound, and do the data support the conclusions?

Reviewer #1: Yes

Reviewer #2: Yes

2. Has the statistical analysis been performed appropriately and rigorously?

Reviewer #1: Yes

Reviewer #2: Yes

3. Have the authors made all data underlying the findings in their manuscript fully available?

Reviewer #1: Yes

Reviewer #2: No

 The data availability has been clarified in the online submission form.

4. Is the manuscript presented in an intelligible fashion and written in standard English?

Reviewer #1: Yes

Reviewer #2: Yes

5. Review Comments to the Author

Reviewer #1: I will focus on methods and reporting. The abstract is well written and balanced. The use of the ICER is appropriate as far as I can tell. Methods are appropriate. Graphs are informative.

Major

1) the sample size, for which the authors cannot do much. But they can tone down the certainty of their findings and conclusions, especialyl in the abstract.

The abstract has been modified.

2) Linked to that there is no power calculation. Was there for the original study? It is worth saying that there was, to detect what (if there was), and then go on to say that you are underpowered to detect anything but very large effects. A post-hoc power calculation is meaningless.

The power calculation has been clarified in the method section and the small study size mentioned as a limitation in the discussion section.

2) missing data - why is a complete case approach selected? Why don't the authors use multiple imputation, for example?

All imputation methods basically invent data in a more or less true way and only works under the assumption that data are missing at random. Multiple imputation would be the preferred way to impute data. However, because we did not have much partially missing data, we decided that it would be most transparent to only use data that was complete rather than use multiple imputation to impute a few more data points (113 were analyzed as complete cases out of 140 included patients). A clarifying note has been added in the methods section.

Minor

1) Introduction was a bit short and did not give a comprehensive picture of the issue.

This is a matter of personal preferences. We think we have defined the need for the study: equal clinical effects of different surgical procedures and a lack of previous studies. We think that it is preferable to keep the introduction short. No changes have been performed. 

2) I appreciate this is an old trial and the information is avaialble elsewhere, but the authors could expand a bit on the randomisation process at least since it is quite important.

A description of the randomization process has been added to the methods section. 

Reviewer #2: This is a generally well-conducted study, and the authors' interpretations and conclusions are supported by the results, although should perhaps be softened somewhat given the wide uncertainty evident in the cost-effectiveness results.

The conclusions have been modified to a more moderate wording in the abstract and conclusion sections. 

The reporting is generally clear, although some minor grammatical issues should be addressed by careful language/copy-editing throughout.

We are not native English speakers and would appreciate identification of grammatical issues by the editorial process.

I have a few further concerns that should be addressed before the manuscript is suitable for publication:

1. The cost-perspective is probably best described as a partial societal perspective, as the only non-healthcare costs considered are production loss due to sick leave (i.e. many other societal costs, such as other out-of-pocket patient costs, caregiver time, travel, etc. are not included)

The description of the perspective has been modified as requested. 

2. Reporting a health care sector perspective in additional to the (partial) societal perspective would be useful (e.g. the dual reference cases recommended by the 2nd panel on cost-effectiveness in health & medicine). This would likely also give more precise results, given how much the productivity losses contribute to the overall uncertainty

3. Would be good to see a completed CHEERS checklist to ensure the manuscript conforms to reporting guidelines for health economic evaluations

A CHEERS checklist has been added as supporting information. 

4. The authors have stated that 'All relevant data are within the manuscript and its Supporting Information files.' If the complete data have been uploaded as Supporting Information, I do not seem to have access to them as a reviewer, and there is no list of supporting information included

The statement has been modified. All data is available upon request from the corresponding author.

5. There are a lot of figures & tables included, some of which could probably be moved to an Appendix (e.g. Table 4) or combined (e.g. Figures 4 & 5)

We think that figures and tables are valuable to make the results clear to our readers. If requested by the editorial office, we will change relevant figures / tables to appendices. 

6. PLOS authors have the option to publish the peer review history of their article (what does this mean?). If published, this will include your full peer review and any attached files.

Do you want your identity to be public for this peer review? For information about this choice, including consent withdrawal, please see our Privacy Policy.

Reviewer #1: No

Reviewer #2: No

All figures have been uploaded through your PACE system, and are provided as TIF files.

---

## [Decision Letter · Decision Letter 1]

26 Aug 2020

PONE-D-20-09180R1

Volar Locking Plate versus External Fixation for Unstable Dorsally Displaced Distal Radius Fractures – A 3-year Cost-Utility Analysis

PLOS ONE

Dear Dr. Mellstrand Navarro,

Thank you for submitting your manuscript to PLOS ONE. After careful consideration, we feel that it has merit but does not fully meet PLOS ONE’s publication criteria as it currently stands. Therefore, we invite you to submit a revised version of the manuscript that addresses the points raised during the review process.

We look forward to receiving your revised manuscript.

Kind regards,

Daniel Ribeiro

Academic Editor

PLOS ONE

Additional Editor Comments (if provided):

Thank you for your revised manuscript. Can I please ask you to address the comments raised by the reviewers? 

Reviewers' comments:

Reviewer's Responses to Questions

**Comments to the Author**

1. If the authors have adequately addressed your comments raised in a previous round of review and you feel that this manuscript is now acceptable for publication, you may indicate that here to bypass the “Comments to the Author” section, enter your conflict of interest statement in the “Confidential to Editor” section, and submit your "Accept" recommendation.

Reviewer #1: All comments have been addressed

Reviewer #2: (No Response)

2. Is the manuscript technically sound, and do the data support the conclusions?

Reviewer #1: Partly

Reviewer #2: Yes

3. Has the statistical analysis been performed appropriately and rigorously? 

Reviewer #1: Yes

Reviewer #2: Yes

4. Have the authors made all data underlying the findings in their manuscript fully available?

Reviewer #1: Yes

Reviewer #2: Yes

5. Is the manuscript presented in an intelligible fashion and written in standard English?

Reviewer #1: Yes

Reviewer #2: Yes

6. Review Comments to the Author

Reviewer #1: I am happy with the authors' responses in general. The argument against multiple imputation is rather weak, especially considering the small sample size. So including another 20 people or so in small study like this would be quite beneficial I think. Yes multiple imputation invents data out of thin air and it comes with assumptions, but it is still the best approach irrespective of the underlying mechanism: https://bmcmedresmethodol.biomedcentral.com/articles/10.1186/s12874-016-0281-5

Reviewer #2: The responses the authors have provided are fine, but there does not appear to be any response or action taken in regard to my comment #2.

Reporting both a health system and societal perspective would be in line with guidelines for the conduct of cost-effectiveness analyses (e.g. Sanders et al., JAMA 2016, doi:10.1001/jama.2016.12195), and the health system-only perspective would likely reduce some of the uncertainty associated with the estimation of productivity costs. If the authors do not believe this analysis is required, this should be justified (but since they have the data required to do this, it seems an easy way to improve the reporting & interpretability of the paper).

7. PLOS authors have the option to publish the peer review history of their article (what does this mean?). If published, this will include your full peer review and any attached files.

Reviewer #1: No

Reviewer #2: No

---

## [Author Response · Author response to Decision Letter 1]

4 Sep 2020

Stockholm 2 September 2020

Response to reviewers 

Rebuttal letter 

Thank you for your response regarding our revision of 

PONE-D-20-09180R1

Volar Locking Plate versus External Fixation for Unstable Dorsally Displaced Distal Radius Fractures – A 3-year Cost-Utility Analysis

PLOS ONE

Please see below our answers to reviewers’ comments:

Additional Editor Comments (if provided):

Thank you for your revised manuscript. Can I please ask you to address the comments raised by the reviewers? 

Reviewers' comments:

Reviewer's Responses to Questions

Comments to the Author

6. Review Comments to the Author

Reviewer #1: I am happy with the authors' responses in general. The argument against multiple imputation is rather weak, especially considering the small sample size. So including another 20 people or so in small study like this would be quite beneficial I think. Yes multiple imputation invents data out of thin air and it comes with assumptions, but it is still the best approach irrespective of the underlying mechanism: https://bmcmedresmethodol.biomedcentral.com/articles/10.1186/s12874-016-0281-5

We understand your point of view. A larger sample would have been optimal. Since our study population size is limited by our RCT patients (out of which 113/140, i e 81% were analyzed) we still consider our choice of complete case analysis be adequate. 

Reviewer #2: The responses the authors have provided are fine, but there does not appear to be any response or action taken in regard to my comment #2.

Reporting both a health system and societal perspective would be in line with guidelines for the conduct of cost-effectiveness analyses (e.g. Sanders et al., JAMA 2016, doi:10.1001/jama.2016.12195), and the health system-only perspective would likely reduce some of the uncertainty associated with the estimation of productivity costs. If the authors do not believe this analysis is required, this should be justified (but since they have the data required to do this, it seems an easy way to improve the reporting & interpretability of the paper).

. 2. Reporting a health care sector perspective in additional to the (partial) societal perspective would be useful (e.g. the dual reference cases recommended by the 2nd panel on cost-effectiveness in health & medicine). This would likely also give more precise results, given how much the productivity losses contribute to the overall uncertainty

Thank you for your suggestion! We agree that this would increase transparency and improve the paper. We have now clarified the differences between the health care and (partial) societal perspective by adding information in table 7, where costs at 1 and 3 years, plus ICER for the respective perspectives have been presented. The total costs for both perspectives are also described in table 5, where standard deviations are presented. We have modified the methods section and discussion in line with your comments. We hope you find these changes satisfying.

When editing this revision we identified a calculation error: in table 7 the ICER for the one year perspective was wrong, and it has been corrected. The erroroneous number was not used in any other calculation or analyses in this paper.

---

## [Decision Letter · Decision Letter 2]

25 Sep 2020

Volar Locking Plate versus External Fixation for Unstable Dorsally Displaced Distal Radius Fractures – A 3-year Cost-Utility Analysis

PONE-D-20-09180R2

Dear Dr. Mellstrand Navarro,

We’re pleased to inform you that your manuscript has been judged scientifically suitable for publication and will be formally accepted for publication once it meets all outstanding technical requirements.

Kind regards,

Daniel Ribeiro

Academic Editor

PLOS ONE

Additional Editor Comments (optional):

Reviewers' comments:

Reviewer's Responses to Questions

**Comments to the Author**

1. If the authors have adequately addressed your comments raised in a previous round of review and you feel that this manuscript is now acceptable for publication, you may indicate that here to bypass the “Comments to the Author” section, enter your conflict of interest statement in the “Confidential to Editor” section, and submit your "Accept" recommendation.

Reviewer #2: All comments have been addressed

2. Is the manuscript technically sound, and do the data support the conclusions?

Reviewer #2: (No Response)

3. Has the statistical analysis been performed appropriately and rigorously? 

Reviewer #2: (No Response)

4. Have the authors made all data underlying the findings in their manuscript fully available?

Reviewer #2: (No Response)

5. Is the manuscript presented in an intelligible fashion and written in standard English?

Reviewer #2: (No Response)

6. Review Comments to the Author

Reviewer #2: (No Response)

7. PLOS authors have the option to publish the peer review history of their article (what does this mean?). If published, this will include your full peer review and any attached files.

Reviewer #2: No

---

## [Editor Report · Acceptance letter]

29 Sep 2020

PONE-D-20-09180R2 

Volar Locking Plate versus External Fixation for Unstable Dorsally Displaced Distal Radius Fractures – A 3-year Cost-Utility Analysis 

Dear Dr. Mellstrand Navarro:

I'm pleased to inform you that your manuscript has been deemed suitable for publication in PLOS ONE. Congratulations! Your manuscript is now with our production department. 

Kind regards, 

on behalf of

Dr. Daniel Ribeiro 

Academic Editor

PLOS ONE